

# Evaluation of waterlogging tolerance and responses of protective enzymes to waterlogging stress in pumpkin

Zhenwei Liu[1,2], Dandan Qiao[1,2], Zhenyu Liu[1,2], Pengwei Wang[1,2], Li Sun[1,2] and Xinzheng Li[1,2]

[1] College of Horticulture and Landscape Architecture, Henan Institute of Science and Technology, Xinxiang, Henan, PR China
[2] Henan Provincial Research Center for Horticultural Plant Resource Utilization and Germplasm Innovation Engineering, Henan, Xinxiang, China

Corresponding author
Zhenwei Liu, sunli0001977@126.com

## ABSTRACT

Waterlogging caused by short and severe, or prolonged precipitation can be attributed to global warming. Pumpkin plants are drought-tolerant but not tolerate to waterlogging stress. Under frequent rain and waterlogging conditions, the production of pumpkins is of lower quality, sometimes rotten, and harvest failure occurs in severe cases. Therefore, it is of great significance to assess the waterlogging tolerance mechanism of pumpkin plants. In this study, 10 novel pumpkin varieties from Baimi series were used. The waterlogging tolerance level of pumpkin plants was evaluated by measuring waterlogging tolerance coefficient of biomass and physiological indices using waterlogging stress simulation method. The criteria to evaluate the waterlogging tolerance capacities of pumpkin plants were also explored. Using principal component and membership function analysis, waterlogging tolerance levels of the pumpkin varieties were ranked as follows: Baimi No. 10>; Baimi No. 5>; Baimi No. 1>; Baimi No. 2>; Baimi No. 3>; Baimi No. 7>; Baimi No. 9>; Baimi No. 6>; Baimi No. 4>; Baimi No. 8. Based on the results, Baimi No. 10 was identified with strong waterlogging tolerance and Baimi No. 8 with weak waterlogging tolerance. The responses of malondialdehyde (MDA), proline, key enzymes responsible for anaerobic respiration, and antioxidant enzymes to waterlogging stress were studied in pumpkin plants. The relative expression levels of related genes were determined using real-time fluorescence quantitative PCR technique. The aim of our study was to assess the waterlogging tolerance mechanism of pumpkin plants, thus laying a theoretical foundation for breeding waterlogging-tolerant varieties in the future. After flooding stress treatment, the antioxidant enzyme activities, contents of proline and alcohol dehydrogenases of Baimi No. 10 and Baimi No. 8 displayed an increase followed by a decrease. All indices of Baimi No. 10 were higher than Baimi No. 8. MDA contents gradually increased, with the content being higher in Baimi No. 8 than Baimi No. 10. The activities of pyruvate decarboxylases (PDCs) in Baimi No. 8 and Baimi No. 10 exhibited a decrease initially, followed by an increase, and then a decrease again. The PDC activity in Baimi No. 8 was generally higher than Baimi No. 10. The relative expression levels of genes encoding superoxide dismutase, peroxidase, catalase, and ascorbate peroxidase were consistent with their corresponding enzyme activities. During the early stage of flooding stress, pumpkin plants waterlogging tolerance was improved by enhancing the expression

levels of antioxidant enzyme encoding genes and increasing the antioxidant enzyme activities.

## INTRODUCTION

In recent years, waterlogging has increased significantly around the globe due to the EI Niño phenomenon (*Tang, Xu & Fang, 1998*). Under waterlogging stress, the normal growth of crops is hindered, resulting in reduced yield or even harvest failure in severe cases (*Tian, 2019*). Pumpkins (*Cucurbita moschata (Duch. ex Lam.) Duch.*) are drought-tolerant crops that are very sensitive to flooding. Frequent rain and waterlogging result in the deterioration of pumpkin quality, rotten melons, and loss of harvest. It not only affects the quality and yield of pumpkins, but also seriously impacts agricultural production and farmers' income. Waterlogging tolerance of crops is a complex quantitative trait that is influenced by multiple factors and the mechanisms also vary with different crops. After flooding, various morphological, physiological, and biochemical indices of crops can be used as an evaluation indicator to measure their waterlogging tolerance (*Shi & Zhou, 2006*). Based on the phenotypic changes of chrysanthemum during flooding, *Yin et al. (2009)* established an evaluation system to measure waterlogging resistance for *chrysanthemum* using the appearance and morphological indicators (*e.g.*, leaf color). At present, research on the waterlogging tolerance of vegetables mostly focuses only on a few crops, such as eggplant, cucumber, and pepper, and there are few research reports regarding the waterlogging tolerance of pumpkins. In this study, we used the multi-factor membership function to evaluate the waterlogging tolerance of pumpkin and conducted principal component analysis (PCA) to enhance the reliability of our evaluation method (*Yang et al., 2016*; *Qi et al., 2011*; *Zheng et al., 2015*; *Li, 2007*).

Under normal growth conditions, the production and scavenging of reactive oxygen species (ROS) in crops cell are in a dynamic equilibrium equilibrium state, this balance is disrupted when crops encounter stress stimuli,it leads to ROS production and metabolism disorders in crops. The antioxidant enzyme system is an important system for scavenging ROS in plants, and it can resist the toxic by-products produced from stress-induced metabolism. Superoxide dismutase (SOD), peroxidase (POD), catalase (CAT), and ascorbic acid peroxidase (APX) are key enzymes in the antioxidant system (*Imahori, Takemura & Bai, 2008*; *Miller et al., 2010*). The APX activity of Solanaceae plants increases significantly under waterlogging stress, which plays a key role in the scavenging mechanism of $H_2O_2$ (*Lin, 2004*). After the watermelon seedlings were induced with waterlogging stress in the leaves, a sequence of events occurred, including the inhibition of SOD activity, the increase of ROS levels, and an increase in SOD and POD activities to scavenge ROS. With the increase in stress, ROS levels further increased, causing membrane lipid peroxidation or delipidation of membrane lipids, which destroyed the membrane structure. Such high

ROS levels also led to the accumulation of malondialdehyde (MDA), caused a decrease in the activity of protective enzymes, and damaged the plasma membrane (*Liu et al., 2016*). After stress treatment, the MDA content of peony leaves increased significantly, and a greater increase was observed with a stronger stress level and longer treatment time (*Wang, 2015*). The MDA content of bitter gourd increased significantly after 4 days of flooding, indicating that the degree of waterlogging was directly proportional to the accumulation of MDA (*Zhu & Zhao, 2016*). Under waterlogging stress, osmotic regulators have certain effects on ROS scavenging. For example, the increase in proline content is associated with the effective scavenging of ROS (*Jiang, 1999*). After 10 days of flooding, the proline content in cucumber was significantly higher than that of the control, with an increase of 58.9% (*Barickman, Simpson & Sams, 2019*).

Soil hypoxia is caused by waterlogging stress, which inhibits the aerobic respiration pathway of roots, increases the activity of anaerobic respiration enzymes, and strengthens the anaerobic respiration pathway. Alcohol dehydrogenase (ADH) and pyruvate decarboxylase (PDC) are key enzymes in the anaerobic respiration pathways (*Baileyserres & Chang, 2005*). ADH-deficient transgenic plants were more sensitive to waterlogging, indicating that ethanol fermentation plays an important role in the plants mechanism of waterlogging tolerance (*Thomas, Guerreiro & Sodek, 2005*). Additionally, the activities of ADH and PDC were accelerated, and the concentrations of ethanol, acetaldehyde, and lactic acid were increased after the cucumber was submerged in water for 48 h (*Xu et al., 2014*).

Different crops exhibit distinct resistance levels during waterlogging stress, and the expression of related genes is greatly associated with genotype. Under adverse stress conditions, some related genes are either induced or silenced, and their products resist the environmental stresses causing morphological and physiological changes in plants that affect their normal growth (*Zhang et al., 2011*; *Long, 2019*). Since real-time fluorescence quantitative polymerase chain raction (RT-qPCR) was invented, it has widely been used in various fields and is one of the important techniques to quantitatively detect gene expression (*Wang et al., 2013*; *Tian et al., 2015*). *Xu (2016)* used RT-qPCR technology to measure the relative expression of waterlogging tolerance-related genes in tea plants.

The characteristics of different varieties of the same crop vary greatly, including waterlogging tolerance level. Different stress levels and stress durations have different effects on crops. Therefore, it is of great significance to study the waterlogging tolerance of different varieties of pumpkin. At present, there is no recognized standard to evaluate the waterlogging tolerance of melons. In order to meet the needs of market structure and improve the competitiveness of Chinese pumpkin, research group has selected and bred a new pumpkin variety of Baimi series with strong growth potential and excellent quality in the early more than ten years of work. The mother and father are both obtained by self-crossing and purification of single plant of high generation. In this study, pumpkin varieties of the Baimi seriesIn this study, pumpkin varieties of the Baimi series were used, and biomass and physiological indices were determined using waterlogging stress simulation method. PCA and membership function analyses were used to identify waterlogging-tolerant varieties of pumpkin and an evaluation method was devised to assess their

waterlogging tolerance. Waterlogging simulation was used to measure the stress response of identified varieties with strong or weak waterlogging tolerance, including changes in MDA content, proline content, and activities of key enzymes for anaerobic respiration and antioxidant enzymes under waterlogging stress. The relative expression levels of related genes were determined using RT-qPCR technique.The response of pumpkin with different waterlogging tolerance to waterlogging stress was studied. This study aimed to explore the mechanisms of waterlogging tolerance in pumpkins, thus laying a theoretical foundation for breeding novel waterlogging-resistant pumpkin varieties.

## MATERIALS AND METHODS

### Experimental materials
The experiment was conducted on 10 pumpkin varieties, including Baimi No. 1, Baimi No. 2, Baimi No. 3, Baimi No. 4, Baimi No. 5, Baimi No. 6, Baimi No. 7, Baimi No. 8, Baimi No. 9, and Baimi No.10. All varieties were provided by the Henan Institute of Science and Technology, Henan, China.

### Experimental design
Waterlogging stress simulations were carried out in the laboratory of the School of Horticulture and Landscape Architecture, Henan University of Science and Technology, from June to July and October to December, 2021. The experiment consisted of two treatments—under conventional (control) and flooded conditions, with a randomized block design and three replicates for each treatment.

A total of 50 seeds (fully developed, no pests, and of similar shape) of each of the 10 tested varieties were sown in nutrient pots after germination through soaking.The seeds were sown in a nutritive pot with a special seedling substrate, Light, temperature and humidity were under conventional management. When the seedlings developed two full leaves and one terminal shoot, the double-pot method (*Liu et al., 2020*) was used under flood stress treatment. The water surface was 2–3 cm higher than the substrate, and plants were watered every day. The control group was managed using standard practices applied in the area. On the 7th day of stress treatment under flooding, leaves and root were collected to measure biomass, relative chlorophyll content, antioxidant enzyme activity, and MDA content.

Baimi No. 10 (strong waterlogging tolerance) and Baimi No. 8 (weak waterlogging tolerance) varieties were selected for the further experiment, and the treatment method remained the same as above. Leaves and roots were taken at 0 (control), 1, 3, 5, and 7d after the treatment. Antioxidant enzyme activity, MDA and proline content, activities of key enzymes responsible for anaerobic respiration and antioxidant enzymes were determined. The relative expression of related genes was determined using real-time fluorescence quantitative PCR technique.

### Measurement methods
The relative content of chlorophyll was determined using a portable chlorophyll instrument, soil and plant analyzer development (SPAD)-502. The SOD and POD
activities were measured by the nitrogen blue tetrazolium photoreduction and guaiacol methods, respectively. MDA content was determined by the thiobarbituric acid method. Proline (PRO) content was determined by acid ninhydrin method. Ultraviolet (UV) spectrophotometry was used to measure CAT, APX, ADH, and PDC activities (*Li, 2000*). The detection kits to measure all these indices were purchased from Beijing Solarbio Technology Co., Ltd. Take SOD for example: 0.1 g pumpkin leaves were weighed and one mL extract was added to make ice bath homogenate. Centrifuged at 4 °C for 10 min, 8,000 rpm, then corresponding reagents were added in turn according to the requirements of the kit. Thoroughly mixed and placed in 37 °C water bath for 30 min. The absorption value A was measured at 560 nm. Percentage inhibition = [(A1$_{blank}$-A2$_{blank}$)- (A$_{measure}$-A$_{contrast}$)] ÷ (A1$_{blank}$-A2$_{blank}$) ×100%. SOD activity (U/g) = [percentage inhibition ÷(1-percentage inhibition) ×V$_{RT}$] ÷ (W ×V$_S$÷V$_{ET}$) × F. V$_{RT}$: Total volume of reaction, 0.2 mL; V$_S$: sample volume, 0.02 mL; V$_{ET}$: Total volume of extract; W: sample weight, g; F: Sample dilution ratio.

RNA extraction and cDNA synthesis: Total RNA was extracted from plant leaves following the instructions (MiniBEST Plant RNA Extraction Kit, Takara Bio Inc., Shiga, Japan) mentioned in the RNA extraction kit. NanoDrop 2000 was used and the absorption value A was measured at 260 nm to detect the concentration and purity of the total RNA. Qualified and quantified total RNA was reverse transcribed into cDNA using Takara Bio Inc. PrimeScript™ II 1st Strand cDNA Synthesis Kit.Reverse transcription reaction system 1 plus RNase free dH2O to 10 μl was mixed and incubated at 65 °C for 5 min, then ice bath quickly. The mixture of reverse transcription reaction system 2 was added into the test tube of ice bath, heated at 42 °C for 30–60 min, heated at 95 °C for 5 min to end the reaction, and put on ice for subsequent experiments or cryopreserved (Table 1).

RT-qPCR: The primer sequences of genes encoding SOD, POD, CAT, and APX were designed according to *Zheng (2020)*, and $\beta$-action was used as the housekeeping gene (Table 2). According to the qPCR reaction system (Table 3), they were added successively, mixed and centrifuged. The reaction solution was placed on the Realtime PCR instrument and PCR was carried out using the following parameters: 95 °C for 5 min, 95 °C for 15 s, 60 °C for 30 s, and a total of 40 cycles. After the reaction, the amplification and melting curves were observed, and the data were analyzed using the $2^{-\Delta\Delta CT}$ (Radio(test/calibrator) $=2^{-[\Delta CT(test)-\Delta CT(calibrator)]}$; $\Delta$Ct =Ct target gene-Ct reference gene; $\Delta \Delta$ Ct = $\Delta$ Ct test-$\Delta$ Ct contrast; Fold Change: $2^{-\Delta\Delta Ct}$) method (*Livak & Schmittgen, 2001*). The relative expressions of the enzyme genes were calculated using the above-mentioned method.

## Statistical analysis

Waterlogging tolerance coefficient (WTC) was calculated using Eq. (1) as follows:

$$WTC = \text{Measured value (treatment)/Measured value (control)} . \tag{1}$$

Based on the calculated WTC values, PCA was performed to obtain comprehensive indices.
The membership function (MF) was calculated using Eq. (2) as follows:

$$MF(X_j) = (X_j - X_{min})/(X_{max} - X_{min}) \tag{2}$$

**Table 1  Retrotranscriptional response system.**

| Reagent | System1 | Reagent | System2 |
|---|---|---|---|
| Template RNA:Total RNA | 2 μg | Template RNA Primer Mixture (come from system1) | 10 μl |
| Primer:Oligo(dT) (50 uM) | 1 μl | 5× Reaction Buffer | 4 μl |
| dNTP Mix (10 mmol/L) | 1 μl | RNase Inhibitor (40 U/μl) | 0.5 μl |
|  |  | MMLV RT (200 U/μl) | 1 μl |
|  |  | RNase free dH$_2$O | Up to 20 μl |

**Table 2  Primer sequences used for real-time fluorescence quantification.**

| The name of the gene | For/backward | Primer sequence (5′to 3′) | The name of the gene | For/backward | Primer sequence (5′to 3′) |
|---|---|---|---|---|---|
| $\beta$-action | F | TCTCTATGCCAGTGGTCGTA | CAT | F | CCGATGCCGCCTAATGTGTTGA |
|  | R | CCTCAGGACAACGGAATC |  | R | CGAACCGCTCTTGCCTATCTGG |
| SOD | F | TCCTTGCCCGACCTCCCTTAT | APX | F | GGCGTTATCCGTCGTAGACACA |
|  | R | GCCTCGTGAAGTTGCTCAAGAG |  | R | TGTGCCAGCGTCATGCCAAG |
| POD | F | TGCTGAACCCTGCCCATGTAGA |  |  |  |
|  | R | GGTGTACCACGGTCGTTCCTCA |  |  |  |

**Table 3  qPCR response system.**

| Reagent | System |
|---|---|
| 2×SYBR real-time PCR premixture | 10 μl |
| PCR specific primer F of 10 uM | 0.4 μl |
| PCR specific primer R of 10 uM | 0.4 μl |
| cDNA | 1 μl |
| RNase free dH$_2$O | Up to 20 μl |

where, $j = 1, 2, \ldots, n$; $X_j$ represents the jth comprehensive index; $X_{min}$ represents the minimum value of the jth comprehensive index; and $X_{max}$ represents the maximum value of the jth comprehensive index.

Weightness (W) was calculated according to the contribution rate of each principal component using Eq. (3):

$$W_j = P_j = \sum_{j=1}^{n} P_j \#(3) = P_j \Bigg/ \sum_{j=1}^{n} P_j \tag{3}$$

where, $j = 1, 2, \ldots, n$; $W_j$ represents the importance of the jth comprehensive index among all the comprehensive indices, and $P_j$ is the contribution rate of the jth comprehensive index of each variety.

Comprehensive evaluation value (CEV) to measure waterlogging tolerance was calculated using Eq. (4):

$$CEV = \sum_{j=1}^{n} \left[ U(X_j) \times W_j \right] \tag{4}$$

where, $j = 1, 2, \ldots,$ n. D is the comprehensive evaluation value of waterlogging tolerance obtained from the comprehensive indices of one tested variety under waterlogging stress conditions.

Statistical analysis of data was performed using DPS 7.55 and SPSS 21.0. Duncan's new multiple range method was used for variance analysis, and GraphPad Prism 8.0 was used for plotting ($P < 0.05$).

## RESULTS

### Evaluation of waterlogging tolerance of Baimi pumpkin varieties

The WTC of each index was calculated using Eq. (1). The results of our study show that the WTC values of fresh aboveground mass were the largest for Baimi No. 2. The WTC values of underground dry mass and fresh mass were the largest for Baimi No. 3. The WTC values of aboveground dry mass were the largest for Baimi No. 5. The WTC values for SOD, POD, and CAT activities were the largest for Baimi No. 10. The WTC of MDA content was the largest for Baimi No. 1. The WTC of the Chlorophyll SPAD value was the largest for Baimi No. 6 (Table 4).

### Principal component analysis(PCA)

PCA was carried out based on the WTCs of nine indices of pumpkin seedlings. The contribution rates of the first three principal components were 46.381%, 28.418%, and 12.032%. The cumulative contribution rate was 86.831%, indicating that the first three principal components explained 86.831% of the total variations.

### Membership function analysis

The membership function value of waterlogging tolerance indices of each pumpkin variety was calculated using Eq. (2). For the first principal component, Baimi No. 10 had the largest $MF(X_1)$, and displayed the highest degree of waterlogging tolerance. It can be seen from $PC_1$ that indices with a large coefficient value showed a positive correlation with waterlogging tolerance. For the second principal component, Baimi No. 1 had the largest $MF(X_2)$ and displayed the highest degree of waterlogging tolerance. The value of Baimi No. 8 was the smallest, and thus the least waterlogging-tolerant variety. It can be seen from $PC_2$ that MDA contents with a large coefficient value had a negative correlation with waterlogging tolerance. Based on this relationship, Baimi No. 1 was the most resistant to waterlogging, which is a false interpretation. For the third principal component, Baimi No. 10 had the largest $MF(X_3)$, and displayed the highest degree of tolerance to waterlogging. The values for Baimi No. 3 were the smallest, indicating it to be the least resistant to waterlogging (Table 5).

### Comprehensive evaluation

The comprehensive evaluation value (CEV) was calculated using Eq. (4). Based on the CEV values, 10 pumpkin varieties were sorted according to their waterlogging tolerance levels as follows: Baimi No. 10>Baimi No. 5>Baimi No. 1>Baimi No. 2>Baimi No. 3>Baimi No. 7>Baimi No. 9>Baimi No. 6>Baimi No. 4>Baimi No. 8 (Table 5).

Liu et al. (2023), *PeerJ*, DOI 10.7717/peerj.1517

**Table 4  Waterlogging resistance coefficient of pumpkin biomass and physiological indexes under the waterlogging.**

| Variety | Waterlogging resistance coefficient=Treatment measured value/contrast measured value | | | | | | | | |
|---|---|---|---|---|---|---|---|---|---|
| | Above ground fresh mass | Underground fresh mass | Above ground dry mass | Underground dry mass | SOD | POD | CAT | MDA | SPAD |
| 1 | 0.938 ± 0.026ab | 0.881 ± 0.004b | 0.902 ± 0..086ab | 0.853 ± 0.028abc | 0.173 ± 0.023f | 1.230 ± 0.035cd | 0.977 ± 0.023cd | 1.645 ± 0.087a | 0.933 ± 0.002a |
| 2 | 0.994 ± 0.002a | 0.858 ± 0.002bc | 0.856 ± 0.060abc | 0.792 ± 0.029c | 0.573 ± 0.023e | 1.313 ± 0.038cd | 1.397 ± 0.017bc | 1.159 ± 0.024c | 0.941 ± 0.003a |
| 3 | 0.965 ± 0.012a | 0.980 ± 0.069a | 0.909 ± 0.011ab | 0.943 ± 0.077a | 1.987 ± 0.026b | 1.633 ± 0.058bc | 0.593 ± 0.020d | 0.685 ± 0.015e | 0.805 ± 0.010c |
| 4 | 0.718 ± 0.009f | 0.784 ± 0.001cd | 0.742 ± 0.027c | 0.766 ± 0.020c | 0.233 ± 0.007f | 0.713 ± 0.024ef | 0.817 ± 0.022cd | 0.841 ± 0.017d | 0.714 ± 0.004e |
| 5 | 0.850 ± 0.004cd | 0.949 ± 0.018ab | 0.961 ± 0.012a | 0.917 ± 0.019ab | 1.130 ± 0.023d | 1.903 ± 0.069b | 1.040 ± 0.046cd | 1.485 ± 0.054b | 0.831 ± 0.005bc |
| 6 | 0.839 ± 0.034d | 0.781 ± 0.040cd | 0.749 ± 0.022c | 0.780 ± 0.017c | 0.660 ± 0.017e | 0.487 ± 0.012f | 0.663 ± 0.044cd | 1.547 ± 0.020ab | 0.954 ± 0.006a |
| 7 | 0.869 ± 0.008cd | 0.896 ± 0.006ab | 0.818 ± 0.032bc | 0.836 ± 0.030bc | 1.693 ± 0.037c | 0.810 ± 0.066ef | 1.893 ± 0.347b | 1.110 ± 0.071c | 0.852 ± 0.022b |
| 8 | 0.780 ± 0.013e | 0.736 ± 0.016d | 0.759 ± 0.007c | 0.772 ± 0.016c | 2.033 ± 0.047b | 0.937 ± 0.007de | 0.467 ± 0.060d | 0.523 ± 0.033f | 0.730 ± 0.013de |
| 9 | 0.943 ± 0.026ab | 0.859 ± 0.027bc | 0.809 ± 0.045bc | 0.853 ± 0.022abc | 1.727 ± 0.038c | 1.757 ± 0.078b | 0.387 ± 0.030d | 0.928 ± 0.038d | 0.845 ± 0.004b |
| 10 | 0.905 ± 0.017bc | 0.915 ± 0.019ab | 0.904 ± 0.033ab | 0.921 ± 0.003ab | 3.093 ± 0.110a | 5.443 ± 0.395a | 5.633 ± 0.673a | 1.130 ± 0.018c | 0.754 ± 0.003d |

**Notes.**

The data are mean ± SD ($n = 3$).

Different lowercase letter after the data in the same column indicate significant differences among varieties ($P < 0.05$).
**Table 5 Membership function value and comprehensive evaluation value and ranking of pumpkin varieties.**

| Variety | PC$_1$ | PC$_2$ | PC$_3$ | MF(X$_1$) | MF(X$_2$) | MF(X$_3$) | CEV | Rank |
|---------|--------|--------|--------|-----------|-----------|-----------|-----|------|
| 1 | −0.016 | 2.239 | 0.493 | 0.424 | 1.000 | 0.685 | 0.649 | 3 |
| 2 | 0.015 | 1.700 | 0.064 | 0.429 | 0.883 | 0.573 | 0.598 | 4 |
| 3 | 1.938 | −0.124 | −2.126 | 0.721 | 0.489 | 0.000 | 0.546 | 5 |
| 4 | −2.802 | −1.534 | 0.160 | 0.000 | 0.185 | 0.598 | 0.143 | 9 |
| 5 | 1.502 | 0.946 | −0.262 | 0.655 | 0.721 | 0.487 | 0.653 | 2 |
| 6 | −2.146 | 1.300 | 1.146 | 0.010 | 0.797 | 0.855 | 0.433 | 8 |
| 7 | −0.046 | −0.014 | −0.023 | 0.419 | 0.513 | 0.550 | 0.468 | 6 |
| 8 | −2.213 | −2.388 | −0.464 | 0.090 | 0.000 | 0.435 | 0.108 | 10 |
| 9 | 0.001 | −0.013 | −0.686 | 0.427 | 0.513 | 0.377 | 0.448 | 7 |
| 10 | 3.768 | −2.113 | 1.699 | 1.000 | 0.059 | 1.000 | 0.692 | 1 |

**Notes.**

The eigenvector expression is:

PC1 = $0.303X1 + 0.420X2 + 0.416X3 + 0.444X4 + 0.300X5 + 0.400X6 + 0.326X7 + 0.066X8 + 0.019X9$;

PC2 = $0.333X1 + 0.172X2 + 0.191X3 + 0.031X4 − 0.405X5 − 0.234X6 − 0.222X7 + 0.577X8 + 0.470X9$;

PC3 = $−0.195X1 − 0.290X2 − 0.158X3 − 0.236X4 − 0.057X5 + 0.359X6 + 0.590X7 + 0.540X8 + 0.159X9$;

PC1, the first principal component; PC2, the second principal component; PC3, the third principal component; X1, Above ground fresh mass; X2, Underground fresh mass; X3, Above ground dry mass; X4, Underground dry mass; X5, SOD; X6, POD; X7, CAT; X8, MDA; X9, Chlorophyll SPAD content.

## Response of pumpkin varieties with different waterlogging tolerance to waterlogging stress

### Response of antioxidant enzymes to waterlogging stress in pumpkin

During waterlogging stress, the activities of SOD, POD, CAT, and APX in Baimi No. 8 and Baimi No. 10 showed first an increasing trend followed by a decrease. The activities of SOD, CAT, and APX for Baimi No. 10 were higher than those of Baimi No. 8, whereas the POD activity of Baimi No. 8 was higher than Baimi No. 10. The activities of SOD, POD, and CAT reached their highest level on the 3rd day, 1.08 times and 1.29 times, 2.24 times and 2.32 times, 3.04 times and 4.19 times higher than the control group, respectively. On the 7th day, the SOD activities of Baimi No. 8 and Baimi No. 10 increased by 8.26% and 29.1%, respectively, as compared to the control group. The POD activity decreased by 0.98 times and 1.16 times as compared to the control values. The CAT activity of Baimi No. 8 decreased by 23.93% as compared to the control, while the CAT activity of Baimi No. 10 increased by 28.30% as compared to the control. The APX activities of these two varieties were always lower than the control and were observed to be lowest on the 7th day. The antioxidant enzyme activities increased during the early stages of waterlogging stress (Figs. 1–4).

### Response of MDA and proline contents to waterlogging stress in pumpkin

The MDA content in the leaves of the two pumpkin varieties gradually increased when the waterlogging stress was prolonged. The MDA contents of Baimi No. 8 and Baimi No. 10 were the highest on the 7th day, being 2.09 and 1.62 times higher than the control, respectively. The MDA content of Baimi No. 8 was always observed to be higher than Baimi No. 10, indicating that Baimi No. 8 had a higher degree of membrane lipid peroxidation, and the damage was greater under waterlogging stress (Fig. 5).

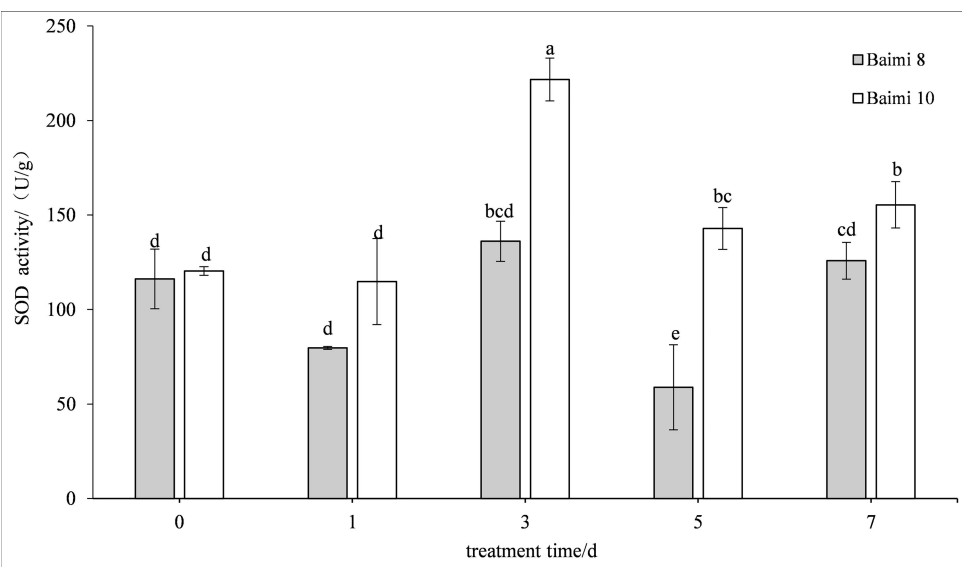

**Figure 1 Response of superoxide dismutase (SOD) to waterlogging stress in pumpkin leaves.** Different letters above columns indicate that the difference of superoxide dismutase activitiesis significant under the water logging ($P < 0.05$). Vertical bars = SD ($n = 3$).

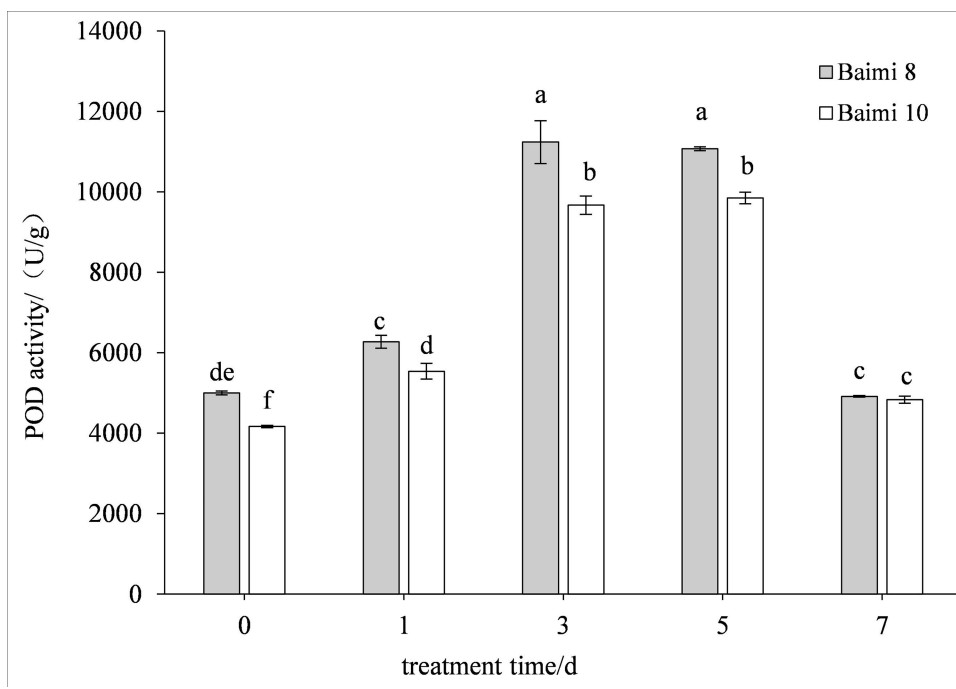

**Figure 2 Response of peroxidase (POD) to waterlogging stress in pumpkin leaves.** Different letters above columns indicate that the difference of peroxidase activities significant under the waterlogging ($P < 0.05$). Vertical bars = SD ($n = 3$).

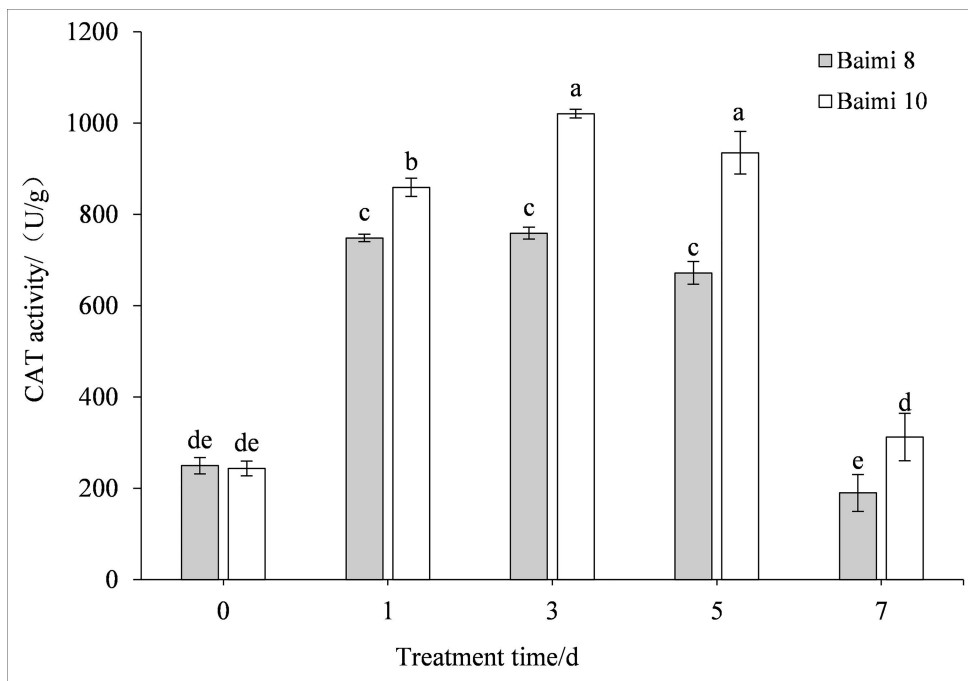

**Figure 3 Response of catalase (CAT) to waterlogging stress in pumpkin leaves.** Different letters above columns indicate that the difference of catalase activities significant under the waterlogging ($P < 0.05$). Vertical bars = SD ($n = 3$).

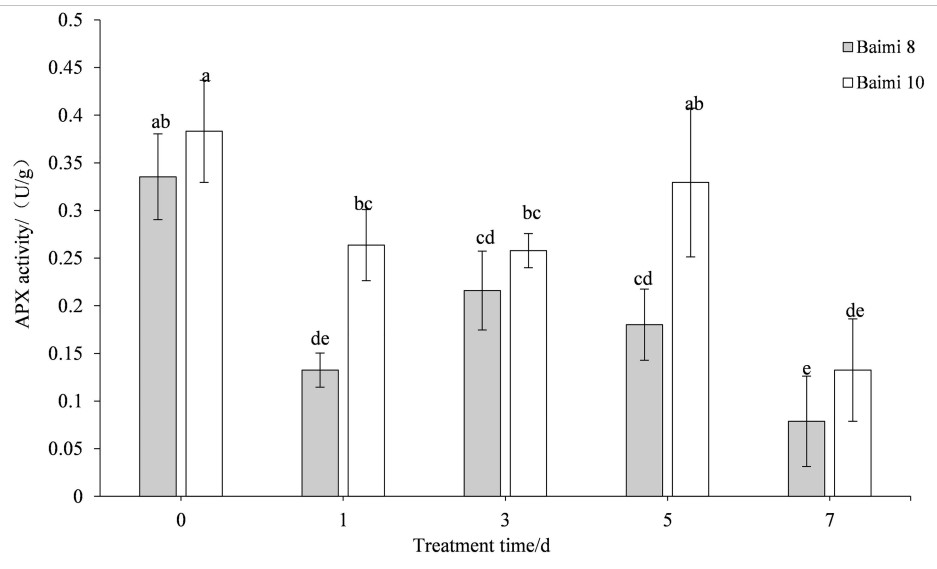

**Figure 4 Response of ascorbic acid (APX) to waterlogging stress in pumpkin leaves.** Different letters above columns indicate that the difference of ascorbic acid activities significant under the water logging ($P < 0.05$). Vertical bars = SD ($n = 3$).

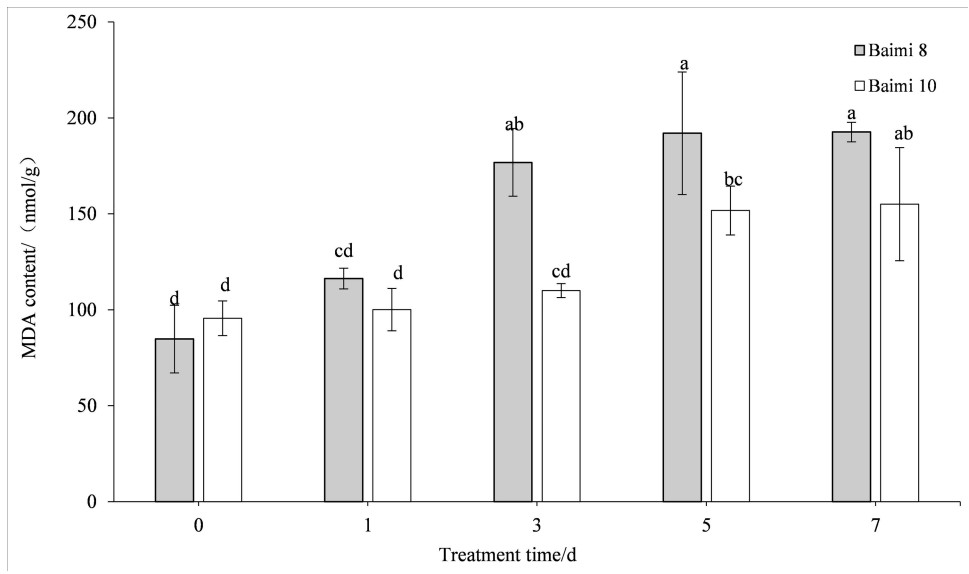

**Figure 5 Response of malondialdehyde (MDA) to waterlogging stress in pumpkin leaves.** Different letters above columns indicate that the difference of the content of malondialdehyde significant under the water logging ($P < 0.05$). Vertical bars = SD ($n = 3$).

During the waterlogging stress, the proline contents of Baimi No. 8 and Baimi No. 10 first displayed an increase followed by a decrease, and reached their peak on the 3rd day, being 1.60 and 4.92 times higher than the control, respectively. During the first phase, the increase in proline content of Baimi No. 10 was 1.49 times higher than the control followed by a phase that exhibited a decrease in which the contents of Baimi No. 10 were 0.61 times higher than the control (Fig. 6).

### Response of key enzymes responsible for anaerobic respiration to waterlogging stress in pumpkin root

The two cultivars first exhibited an increase in their ADH activities followed by a decrease when the waterlogging stress was prolonged. The ADH activity of Baimi No. 10 was higher than Baimi No. 8, which is always lower than the control values. The ADH activity of Baimi No. 10 was observed to be the highest on the 5th day. On the 7th day, the ADH activities of Baimi No. 8 and Baimi No. 10 decreased by 73.33% and 25% than the control, respectively. The PDC activities of Baimi No. 8 and Baimi No. 10 showed a trend that decreased first, rose and then decreased again. The PDC activity of Baimi No. 10 was generally higher than the Baimi No. 8. Baimi No. 8 was always lower than the control values. On the 1st day, the PDC activity of Baimi No. 10 increased 72.09% higher as compared to the control. The PDC activities of the Baimi No. 8 and Baimi No. 10 decreased to their lowest value on the 3rd day, with 58.50% and 56.28% lower than the control, respectively. The activities of key enzymes responsible for anaerobic respiration were higher in the variety with strong waterlogging tolerance as compared to the one with lower tolerance (Figs. 7–8).

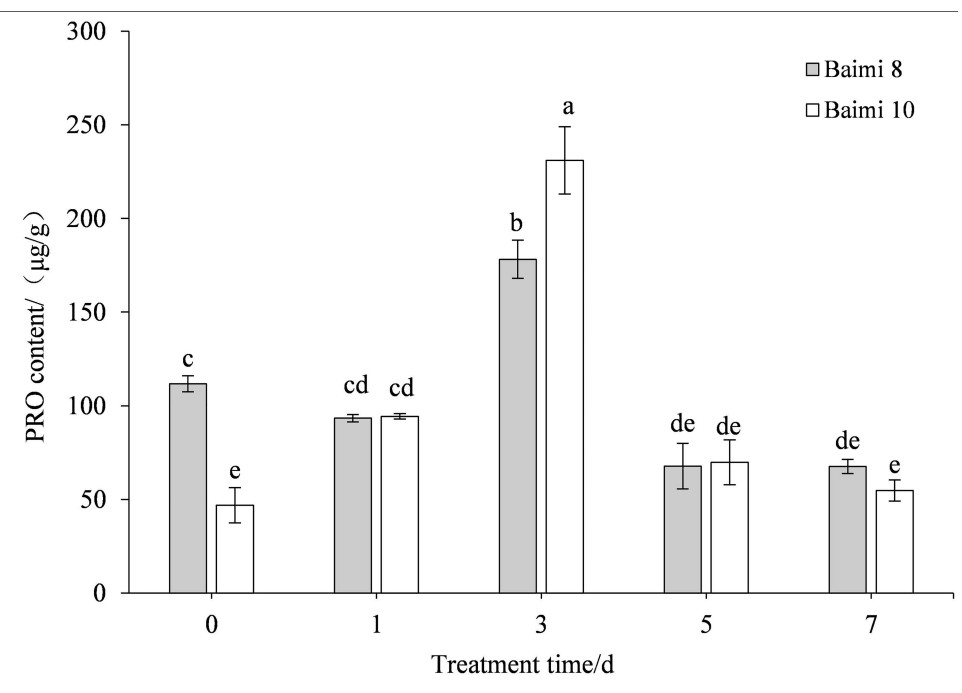

**Figure 6  Response of proline (PRO) to waterlogging stress in pumpkin leaves.** Different letters above columns indicate that the difference of the content of proline significant under the waterlogging ($P <$ 0.05). Vertical bars = SD ($n = 3$).

### Response of pumpkin antioxidant enzyme genes expressions to waterlogging stress

During the waterlogging stress, the expression levels of genes encoding SOD, POD, CAT, and APX in Baimi No. 8 and Baimi No. 10 all exhibited an increase first followed by a decrease. The genes encoding SOD, POD, and CAT were highly expressed on the 3rd day, being 3.05 and 18 times, 3.05 and 2.37 times, 8.87 and 11.56 times than the control, respectively. The expression level of the gene encoding APX in Baimi No. 10 increased to its highest level on the 3rd day, being 7.78 times higher than the control, and in Baimi No. 8 it was the highest on the 5th day. The relative expression levels of genes encoding SOD, CAT, and APX in Baimi No. 10 were higher than the Baimi No. 8, and the relative expression levels of the gene encoding POD in Baimi No. 8 were higher than the Baimi No. 10. The trends of antioxidant enzyme gene expression levels were consistent with their corresponding enzyme activities (Figs. 9–12).

## DISCUSSION

In previous studies, the indices to measure waterlogging tolerance have been screened out in vegetables and mainly focused on growth, physiological, and biochemical indicators. In this study, 10 pumpkin varieties were used to assess the waterlogging tolerance level of pumpkins. PCA and membership function analysis were used to convert nine indices to three independent comprehensive indicators, including biomass, relative chlorophyll

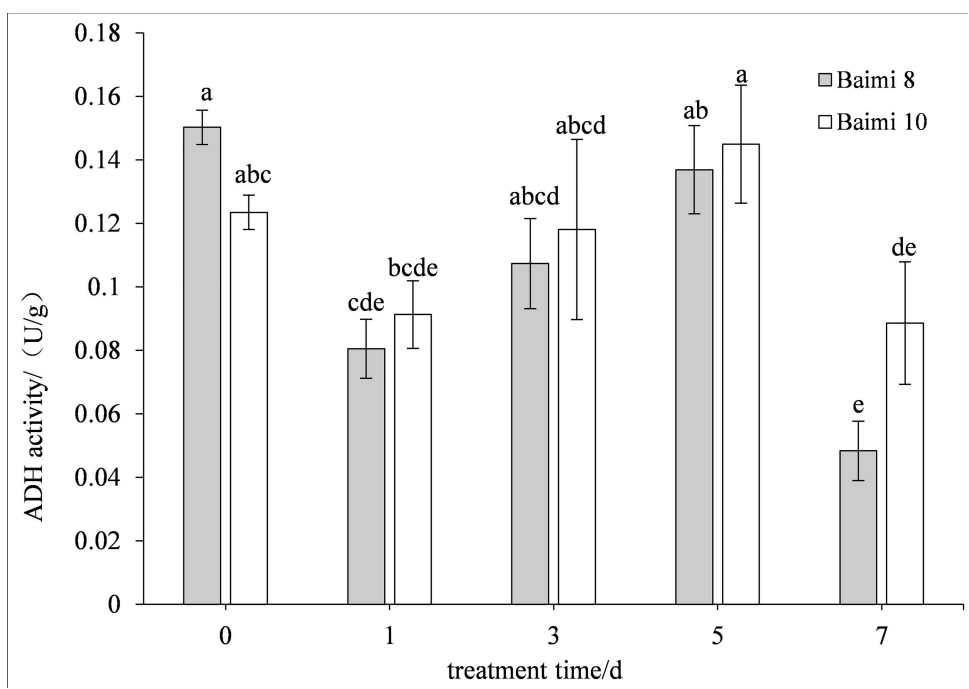

**Figure 7 Response of alcohol dehydrogenase (ADH) to waterlogging stress in pumpkin roots.** Different letters above columns indicate that the difference of alcohol dehydrogenase (ADH) activity significant under the waterlogging ($P < 0.05$). Vertical bars = SD ($n = 3$).

content, antioxidant enzyme activity, and MDA content. The corresponding membership function values were weighted to obtain the comprehensive evaluation value (CEV) of waterlogging tolerance. *Gao et al. (2018)* screened out fresh underground mass, SOD, and MDA content as valid indicators for rapid evaluation of waterlogging tolerance in broccoli seedlings, which suggests that growth indices and antioxidant enzyme activities can be used as an indicator to evaluate waterlogging tolerance.

During the initial stage of waterlogging stress, ROS accumulation in plants was accompanied by improved antioxidant enzyme activities such as SOD, POD, CAT, and APX, protecting plants from ROS damage (*Limón-Pacheco & Gonsebatt, 2008*). This was an important pathway that helps plants cope with all kinds of stress. SOD was considered as first defense against ROS as it acted on superoxide radicals, which were produced in different compartments of the cell and acted as precursor to other ROS (*Alscher, Erturk & Heath, 2002*). APX rapidly reduced $H_2O_2$ to $H_2O$ and $O_2$ through the ascorbate-glutathione cycle (*Mishra et al., 2006*). *Li (2007)* reported that SOD, CAT, and POD activities in cucumber leaves increased three days before waterlogging occurred, and then gradually decreased. *Yang et al. (2014)* found that SOD, POD, and CAT activities in tomato leaves first exhibited an increase followed by a decrease under water stress conditions. In this study, SOD, POD, CAT, and APX activities in pumpkin also displayed an increase first and then a decrease under waterlogging stress, which is consistent with *Li (2007)* and *Yang et al. (2014)*, indicating that short-term waterlogging stress stimulated the antioxidant system of plants,

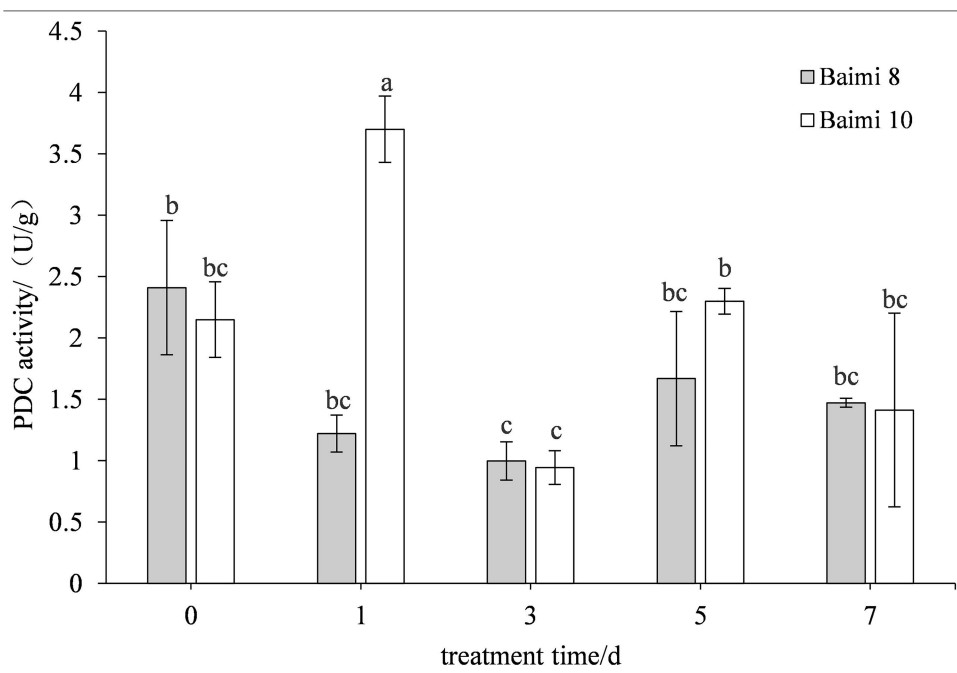

**Figure 8 Response of pyruvate decarboxylase (PDC) to waterlogging stress in pumpkin roots.** Different letters above columns indicate that the difference of pyruvate decarboxylase activity significant under the waterlogging ($P < 0.05$). Vertical bars = SD ($n = 3$).

formed a certain waterlogging resistance mechanism, and maintained a high protective enzyme.However, long-term stress would cause certain damage to the antioxidant system, so that its antioxidant capacity gradually faded. The antioxidant enzyme activity of Baimi No. 10 was higher than the Baimi No. 8, indicating that Baimi No. 10 has an efficient enzymatic scavenging system to regulate the ROS levels. The SOD, POD, CAT, and APX activities in waterlogging-tolerant maize have been reported to enhance to a larger extent as compared to varieties that are sensitive to waterlogging stress (*Wang et al., 2018*).

MDA is an important indicator to measure plant injury under stressed conditions. *Zhu & Zhao (2016)* found that the higher the MDA content, the weaker the waterlogging resistance in bitter gourd. In this study, the MDA content continued to increase when waterlogging stress was prolonged. The MDA content of Baimi No. 8 was higher than Baimi No. 10, and these results were consistent with *Zheng (2020)*, where the MDA content in waterlogging-sensitive watermelon varieties was significantly higher than the waterlogging-resistant varieties. Under stress conditions, the important role of proline may be as a scavenger and penetrant to improve crop growth characteristics and yield (*Khaled Abdelaal et al., 2020*). *Wang (2010)* found that after waterlogging, the content of free proline in melons increased first and then decreased. The proline content in waterlogging-tolerant varieties was significantly higher than the sensitive varieties. In this study, under waterlogging stress, the proline contents of Baimi No. 8 and Baimi No. 10 both followed a similar trend with an increase first and then a decrease. The proline content of Baimi No. 10 was higher

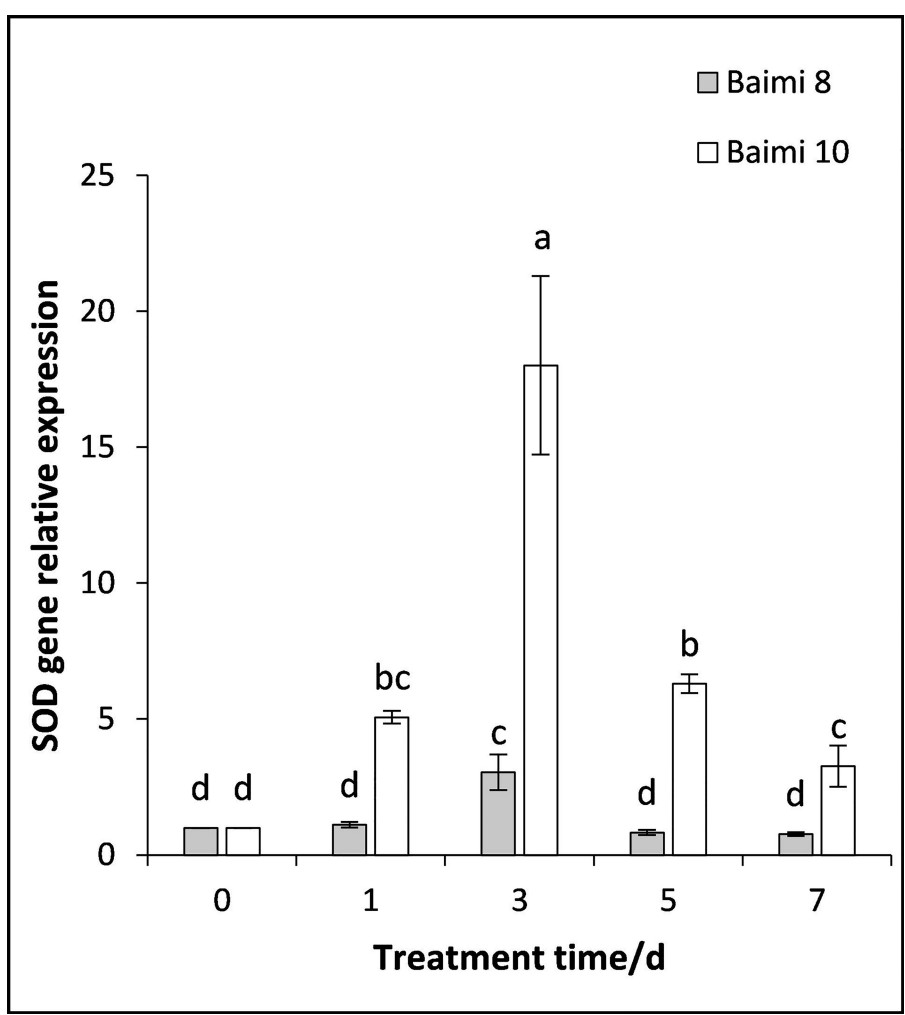

**Figure 9** **Response of gene expression of superoxide dismutase (SOD) to waterlogging stress in pumpkin leaves.** Different letters above columns indicate that the difference of the gene expression of superoxide dismutase significant under the waterlogging ($P < 0.05$). Vertical bars. = SD ($n = 3$).

than Baimi No. 8, indicating that the waterlogging tolerant pumpkin varieties reduced the oxidation of proline to glutamic acid, which made the proline rapid accumulation to reduce the osmotic potential of cells and relieved the damage of waterlogging stress during the early stage (*ALKahtani et al., 2021*).

Plants respond to waterlogging stress by releasing a small amount of energy for their growth through anaerobic respiration and regulating their metabolic pathways. During anaerobic respiration, ADH is the main enzyme that prolongs the survival time of plants through the Pasteur effect under hypoxic conditions (*Chen, Yan & Xiao, 2005*). *Diao et al. (2020)* found that under waterlogging stress, ADH and LDH activities increased in melons, and the ADH activity was higher in 'Cuixi', a variety with slightly higher waterlogging tolerance, as compared to 'Century Honey', a variety with weaker tolerance. In this study, the ADH activities of both Baimi No. 8 and No. 10 displayed and an increasing trend

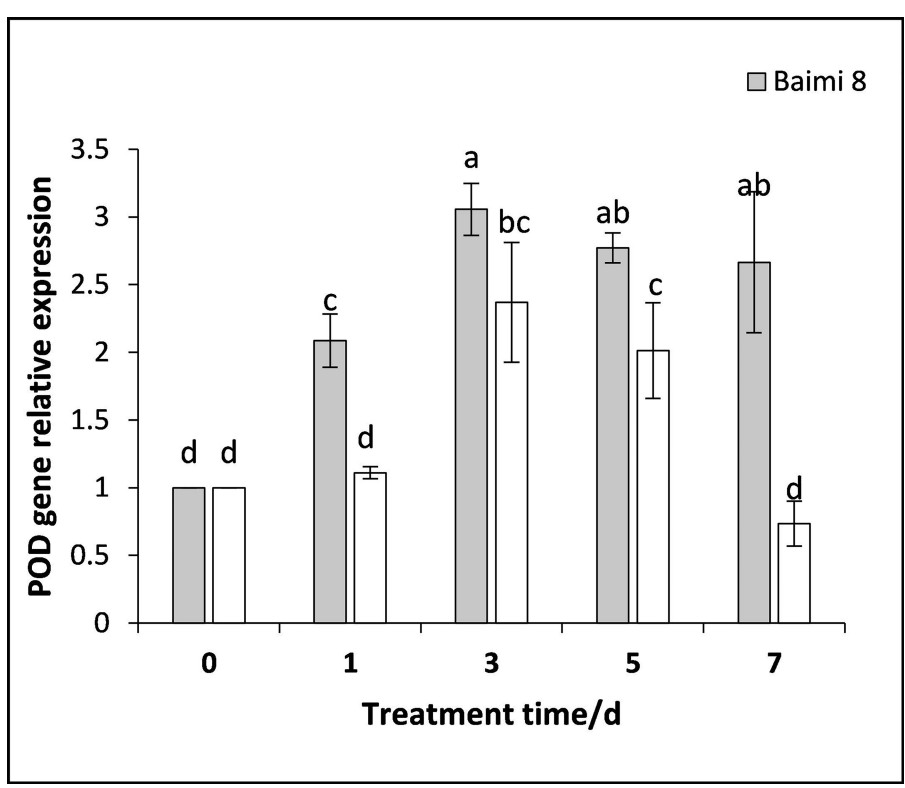

**Figure 10** **Response of gene expression of peroxidase (POD) to waterlogging stress in pumpkin leaves.** Different letters above columns indicate that the difference of the gene expression of peroxidase significant under the waterlogging ($P < 0.05$). Vertical bars = SD ($n = 3$).

followed by a decrease. The ADH activity in Baimi No. 10 was higher than Baimi No. 8, which is consistent with *Diao et al. (2020)*. The PDC activities of Baimi No. 8 and Baimi No. 10 showed a trend that decreased first, followed by an increase and a decrease again. The PDC activity of Baimi No. 10 was generally higher than Baimi No. 8. These results were consistent with *Chen et al. (2007)*, who reported that the PDC activity of the waterlogging tolerant rootstock variety 'Mahali' was higher than the sensitive variety 'Dongbei Shanying'.

*Xu (2016)* assessed the resistance mechanism of tea plants and found that the relative expressions of SOD genes in leaves increased and those of CAT genes in leaves decreased first and then increased. *Chin, Kuo & Lin (2014)* found that in loofah plants, the expression of the gene encoding APX was enhanced with the increase in enzyme activity, resulting in an increased scavenging ability of ROS under flooding stress . In this study, under waterlogging stress, the expression levels of SOD, POD, CAT, and APX encoding genes displayed a trend of an increase first and then a decrease, which was consistent with the trend of their corresponding enzyme activities. In Baimi No. 10, the expression of the APX encoding gene was significantly higher than Baimi No. 8, which is consistent with the findings of *Xia (2015)*. The results showed that in the early stage of stress, the expression of antioxidant enzyme genes was enhanced, which significantly increased the enzyme activity.

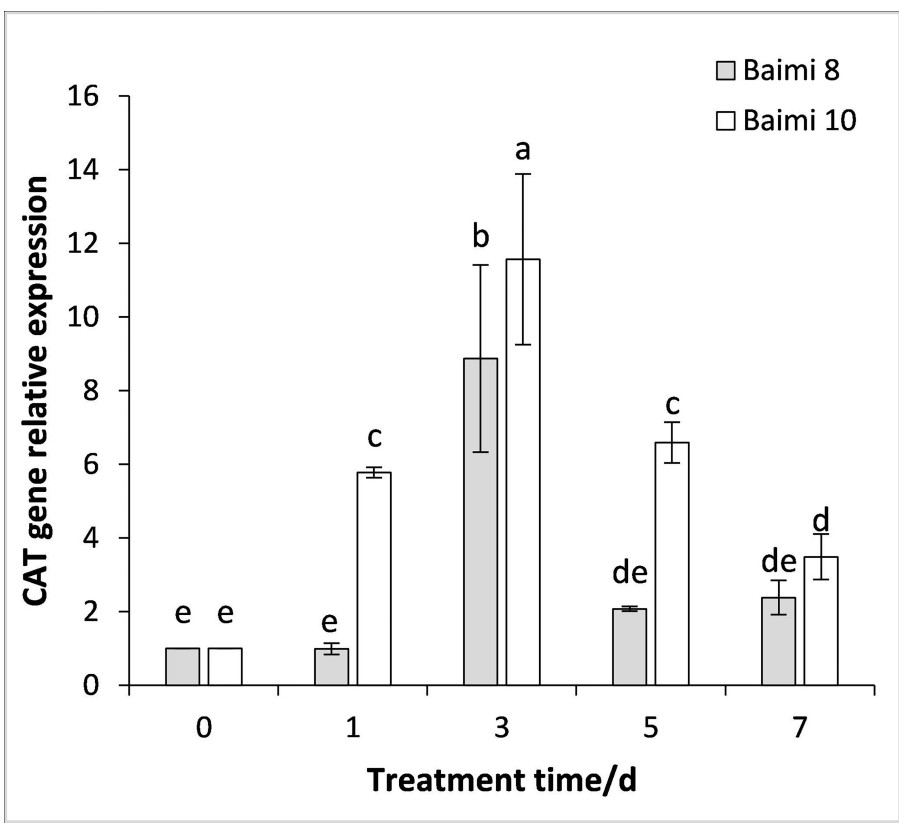

**Figure 11 Response of gene expression of catalase (CAT) to waterlogging stress in pumpkin leaves.** Different letters above columns indicate that the difference of the gene expression of catalase significant under the waterlogging ($P < 0.05$). Vertical bars = SD ($n = 3$).

Consequently, the production and scavenging of ROS can maintain an equilibrium state for a certain period, thereby reducing damage to plants.

## CONCLUSION

In this study, biomass, antioxidant enzyme activity, and relative chlorophyll content of 10 different pumpkin varieties were used as indicators to evaluate their waterlogging tolerance. Through PCA and membership function analysis, we identified the varieties that were the most tolerant to waterlogging. Among 10 tested pumpkin varieties, Baimi No. 10 was the most tolerant and Baimi No. 8 was the least tolerant. Using Baimi No. 10 and Baimi No. 8 varieties, we further studied the responsive activities of MDA, proline, key enzymes responsible for anaerobic respiration, antioxidant enzymes, and the expression of encoding genes to waterlogging stress in pumpkin plants. When Baimi No. 8 was compared with Baimi No. 10, it was found that it may possess a more efficient enzymatic scavenging system to regulate ROS. The MDA content of Baimi No. 8 was higher than that of Baimi No.10, and the degree of lipid peroxidation was higher. The relative expression levels of SOD, POD, CAT, and APX encoding genes were consistent with their corresponding enzyme activities. All of them increased first and then decreased, indicating that the expression of

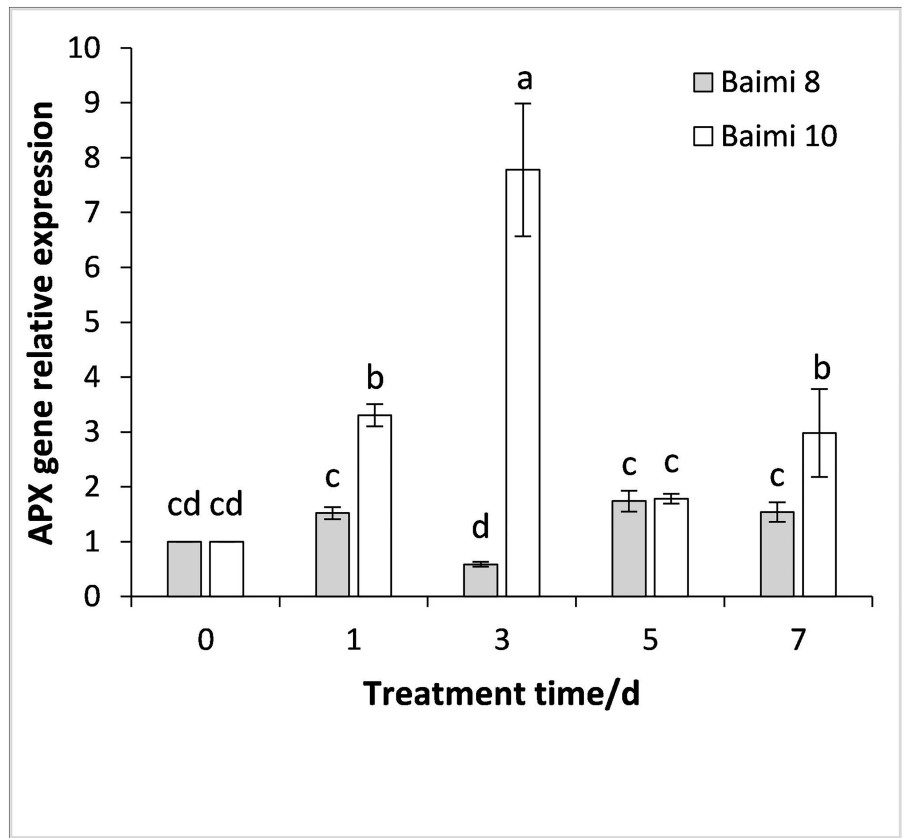

**Figure 12  Response of gene expression of ascorbic acid (APX) to waterlogging stress in pumpkin leaves.** Different letters above columns indicate that the difference of the gene expression of ascorbic acid significant under the waterlogging ($P < 0.05$). Vertical bars = SD ($n = 3$).

antioxidant enzymes in plants was up-regulated by short time of waterlogging stress, and a certain waterlogging tolerance mechanism was formed. However, with the extension of stress time, the antioxidant system was destroyed. In the early stage of waterlogging stress, pumpkin plants may resist waterlogging by enhancing the expressions of antioxidant enzyme-encoding genes and improving enzyme activity. However, after 3 days of stress treatment, the gene expression levels of antioxidant enzymes and their activities decreased. The degree of membrane lipid peroxidation increased, and plant growth was inhibited. The antioxidant enzyme activities and related gene expressions were higher in the variety with strong waterlogging tolerance than in the variety with weak waterlogging tolerance.

## ACKNOWLEDGEMENTS

My deepest gratitude goes first and foremost to Professor Li, for constant encouragement and guidance. Without supporting of his project, this thesis could not have reached its present form. Second, I would like to express my heartfelt gratitude to my students, Qiao Dandan, Yan Xiaowen, et al. who gave me their help in my experiment.

### Funding

The research was supported by the Special funded projects of key research and development and promotion of Henan Province (No. 202102110201). The funders had no role in study design, data collection and analysis, decision to publish, or preparation of the manuscript.

### Grant Disclosures

The following grant information was disclosed by the authors:
Key research and development and promotion of Henan Province: 202102110201.

### Competing Interests

The authors declare there are no competing interests.

### Author Contributions

- Zhenwei Liu conceived and designed the experiments, performed the experiments, analyzed the data, prepared figures and/or tables, authored or reviewed drafts of the article, and approved the final draft.
- Dandan Qiao conceived and designed the experiments, performed the experiments, analyzed the data, prepared figures and/or tables, authored or reviewed drafts of the article, and approved the final draft.
- Zhenyu Liu performed the experiments, analyzed the data, prepared figures and/or tables, and approved the final draft.
- Pengwei Wang performed the experiments, analyzed the data, prepared figures and/or tables, and approved the final draft.
- Li Sun conceived and designed the experiments, analyzed the data, prepared figures and/or tables, and approved the final draft.
- Xinzheng Li conceived and designed the experiments, authored or reviewed drafts of the article, his project supports the experiment, and approved the final draft.

### Data Availability

The raw measurements are available in the Supplemental Files.

### Supplemental Information

Supplemental information for this article can be found online at http://dx.doi.org/10.7717/peerj.15177#supplemental-information.

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
