# Peer review of "Evaluation of waterlogging tolerance and responses of protective enzymes to waterlogging stress in pumpkin"

_PeerJ, doi:10.7717/peerj.15177_

## Round 0.1 · original submission · Major Revisions

Dear Authors

The manuscript cannot be accepted for publication in its current form. It needs a major revision to be reconsidered for publication. The authors are invited to revise the paper considering all the suggestions made by the reviewers. Please note that requested changes are required for publication.

Reviewer 2 has requested that you cite specific references. You may add them if you believe they are especially relevant. However, I do not expect you to include these citations, and if you do not include them, this will not influence my decision.

With Thanks

·

Basic reporting

The manuscript is well written but still needs some minor changes

Experimental design

No comments

Validity of the findings

No comments

Additional comments

I reviewed the paper titled "Evaluation of Waterlogging Tolerance and Responses of Protective Enzymes to Waterlogging Stress in Pumpkin". The authors aimed to explore the mechanisms of waterlogging tolerance in pumpkins, thus laying a theoretical foundation for breeding novel waterlogging-resistant pumpkin varieties. The results are interesting and applicable. The manuscript needs minor changes.
Abstract
1- Line 46,Using principal component …correct to ….Using the principal component
2- Line 53, The aim of our study was to …correct to …. Our study aimed to
Introduction
The introduction section is comprehensive and well written. Please add the scientific name of Pumpkin
1- Line 73, results …correct to …. Result
2- Line 75, farmers …correct to …. farmers'
3- Line 87, relatively …correct to …. relative
4- Line 88, it has the ability to replace with (can)
5- Line 96, resulted in the destruction of replace with (destroyed)
6- Line 99 the stronger replace with (a stronger)
7- Line 127, analysis replace with (analyses)
8- Line 132, The aim of this study was to…. correct to….. This study aimed to
Materials and methods
This section needs more details to enable the other researchers to repeat any experiment (please add more details especially the PCR reactions)
Line 145 with no pests
Line 153, selected for the further experiment
Line 165, Please mention the wave length used for quantification of RNA by NanoDrop
Line 166, please add the conditions of reverse transcription reaction
Line 173 the data were
Results
The results section is well written. I suggest the authors to error bars with standard error to decrease the bar length in all figures
Discussion
Line 276, seedlings , which suggest that correct to seedlings, which suggests that
Line 281, the correct spelling is occurred
Line 281, you have improperly spaced punctuation…remove space after decreased (Please revise the whole manuscript)
Line 310 that decrease
Conclusion
Line 333 a more
Line 336, enzyme-encoding
References
Please unify the style according to the journal instructions; some references included the Doi while the other did not.

·

Basic reporting

no comment

Experimental design

no comment

Validity of the findings

no comment

Additional comments

please follow the comments in the PDF file as follow:
page 2, line 14
page 3, line 1
page 3, lines 36-40
page 4 , line 1
page 4 line 35
page 5, line 4
page 8, lines 6,11,19,27 and 42
please see the pdf version
Best regards

Reviewer 3 ·

Basic reporting

• In the keywords, it is strongly advisable to use suitable words that can aid in finding out the manuscript in current registers or indexes. Strictly avoid the use of title words in the keywords. For Example: Pumpkin, Waterlogging stress
• Line 72, please check and correct the space in statement “severe cases (Tian L 2019)”.
• Line 79, please check and correct the scientific name of the plant “Chrysanthemum”. It should be italic.
• “Under normal growth conditions, the production and scavenging of reactive oxygen species (ROS) in crops are 87 in a relatively equilibrium state, and the balance would be disturbed if stress occurs” line 86-87, rephrase the sentence as it seems to be incorrect.
• “The effects on crop physiology also vary with stress level and duration” Line 123, rephrase the sentence as the meaning was not clear.
• The introduction section should include the little bit information about the pumpkin variety “Baimi”.
• Your introduction needs more detail. Line 132-134, the objective mentioned in the last part of the manuscript need to be more precise as it repeating the title and also mention some sub objective regarding the experiment.

Experimental design

• The authors should mention the nutrient condition they were giving to the seedlings.
• The authors should also select the intermediate waterlogging tolerance variety of Baimi for better analysis of the result.
• The authors should elaborate the methods they had used for antioxidant enzymes activity analysis. The methods was not described with sufficient detail.

Validity of the findings

• The impact and novelty statement was missing from the manuscript.
• Many statements/sentences throughout the manuscript require proper validation and citation with previous studies, which are largely missing in the present manuscript. For example, in lines 298-301, that statement needs proper validation and citation.
• The discussion section was not elaborate as they had not explained the non-significant results like for the SPAD analysis, Baimi 8 and 10 don’t have any significant difference, so it should be explained in the discussion section. The discussion did not provide a specific reasons for the results and also the provided explanation could have been more satisfactory also.
• The reasons for the variation in antioxidant enzyme activities during the different time intervals need to be explained in the discussion section day 3 data has the highest antioxidant enzyme activity as compared to day 7 data.
• In the conclusion section the authors have mentioned the data. For example, lines 333-335 just repeating the results, so the major finding and the appropriate reasons needs to be incorporated in the conclusion part.
• Need to rewrite and incorporate this important concern of reviewer.
• The introduction, Result, and Discussion sections poorly cited with the references and strongly recommended to update and validation with previous studies. Therefore, some of the papers listed below should be considered and cited appropriately in the Introduction, Result, and Discussion section of this Ms, which will certainly upgrade and enhance the Ms.quality significantly.
 Mishra, S., et al. (2006). Lead detoxification by coontail (Ceratophyllumdemersum L.) involves induction of phytochelatins and antioxidant system in response to its accumulation. Chemosphere, 65(6), 1027-1039.
 Agnihotri, A., et al. (2020). Does jasmonic acid regulate photosynthesis, clastogenecity, and phytochelatins in Brassica juncea L. in response to Pb-subcellular distribution?. Chemosphere, 243, 125361.
 Gupta, P., et al. (2022). 24-Epibrassinolide Regulates Functional Components of Nitric Oxide Signalling and Antioxidant Defense Pathways to Alleviate Salinity Stress in Brassica juncea L. cv. Varuna. Journal of Plant Growth Regulation, 1-16.
 Prajapati, P., et al. (2022). Nitric oxide mediated regulation of ascorbate-glutathione pathway alleviates mitotic aberrations and DNA damage in Allium cepa L. under salinity stress. International Journal of Phytoremediation, 1-12.

Additional comments

• Grammatical errors are present, please revise the whole manuscript to remove any possible grammatical and typos errors.
• Error in sentence formation, please revise the whole manuscript to avoid the use of long sentences.
• Please maintain uniformity while in-text citation and referencing. For example: “Chen Q,Guo X, Hu Y,et al.(2007)Effects of waterlogging on anaerobic respiration enzymes and fermentation products in roots of two kind of sweet cherry root stocks.ActaEcologicasinica 27(11):4925-4931” was cited as (Chen et al.2007) while “Chen Y,YanQ,Xiao G(2005) Progresses in research of submergence tolerance in rice.Chinese Agricultural Science Bulletin 21(12):151-153.” was cited as (Chen Y et al.2005).
• References have not been appropriately cited in the Ms.It needs critical revision and updation. For example: “BarickmanTC,Simpson CR, Sams CE(2019)Waterlogging causes early modification in the physiological performance,carotenoids,chlorophylls,Proline and soluble sugars of cucumber plants.Plants 8(6):160.https//doi:10.3390/plants8060160” has DOI while “Chen Q,Guo X, Hu Y,et al.(2007)Effects of waterlogging on anaerobic respiration enzymes and fermentation products in roots of two kind of sweet cherry root stocks.ActaEcologicasinica 27(11):4925-4931.” don’t have DOI.
• Please provide the suitable references for calculating Waterlogging tolerance coefficient (WTC)and for statistical analysis.

---

## Round 0.2 · accepted · Accept

Dear Authors

I am pleased to inform you that after the last round of revision, the manuscript has been improved a lot, and it can be accepted for publication.

Congratulations on the acceptance of your manuscript, and thank you for your
interest in submitting your work to PeerJ.

Best Regards

EDITS LINE NO: / BEFORE / AFTER / [COMMENTS]
LINE 127-131 / Chinese pumpkin, research group has selected and bred a new pumpkin variety of Baimi series with strong growth potential and excellent quality in the early more than ten years of work. The mother and father are both obtained by self-crossing and purification of single plant of high generation. In this study, pumpkin varieties of the Baimi seriesIn this study, pumpkin varieties of the Baimi series were used, and biomass and physiological indices were determined using waterlogging stress simulation method. /
Chinese pumpkin, this research group has selected and bred a new pumpkin variety of the Baimi series with strong growth potential and excellent quality in over ten years of effort. The mother- and father-lines are both obtained by self-crossing and purification of a single plant of high generation. In this study, pumpkin varieties of the Baimi series were used in this study, and biomass and physiological indices were determined using a waterlogging stress simulation method. / [suggested editing]
LINE 152-154: / seedling substrate,Light, temperature and humidity were under conventional management. When the seedlings developed two full leaves and one terminal shoot, the double-pot method (Liu C et al.2020) was used under flood stress treatment /
seedling substrate, with light, temperature and humidity under conventional management. When the seedlings developed two full leaves and one terminal shoot, the double-pot method (Liu C et al.2020) was used for flood stress treatment / [suggested editing]
LINE 159: / experiment / experimentation / [.]

·

Basic reporting

No comment

Experimental design

No comments

Validity of the findings

No comment

Additional comments

The manuscript is now considered an informative piece of research and a helpful share of knowledge about the subject, which is within the scope of the journal.

·

Basic reporting

No comment

Experimental design

No comment

Validity of the findings

No comment

Reviewer 3 ·

Basic reporting

excellent

Experimental design

excellent

Validity of the findings

excellent

Additional comments

excellent